# Dental Implant Navigation System Based on Trinocular Stereo Vision

**DOI:** 10.3390/s22072571

**Published:** 2022-03-27

**Authors:** Songlin Bi, Menghao Wang, Jiaqi Zou, Yonggang Gu, Chao Zhai, Ming Gong

**Affiliations:** 1Department of Precision Machinery and Precision Instrumentation, University of Science and Technology of China, Hefei 230027, China; bisl001@mail.ustc.edu.cn (S.B.); wmh0927@mail.ustc.edu.cn (M.W.); zjq007@mail.ustc.edu.cn (J.Z.); 2Experiment Center of Engineering and Material Science, University of Science and Technology of China, Hefei 230027, China; yggu@ustc.edu.cn (Y.G.); gongming@ustc.edu.cn (M.G.)

**Keywords:** dental implant, trinocular stereo vision, dynamic tracking

## Abstract

Traditional dental implant navigation systems (DINS) based on binocular stereo vision (BSV) have limitations, for example, weak anti-occlusion abilities, as well as problems with feature point mismatching. These shortcomings limit the operators’ operation scope, and the instruments may even cause damage to the adjacent important blood vessels, nerves, and other anatomical structures. Trinocular stereo vision (TSV) is introduced to DINS to improve the accuracy and safety of dental implants in this study. High positioning accuracy is provided by adding cameras. When one of the cameras is blocked, spatial positioning can still be achieved, and doctors can adjust to system tips; thus, the continuity and safety of the surgery is significantly improved. Some key technologies of DINS have also been updated. A bipolar line constraint algorithm based on TSV is proposed to eliminate the feature point mismatching problem. A reference template with active optical markers attached to the jaw measures head movement. A T-type template with active optical markers is used to obtain the position and direction of surgery instruments. The calibration algorithms of endpoint, axis, and drill are proposed for 3D display of the surgical instrument in real time. With the preoperative path planning of implant navigation software, implant surgery can be carried out. Phantom experiments are carried out based on the system to assess the feasibility and accuracy. The results show that the mean entry deviation, exit deviation, and angle deviation are 0.55 mm, 0.88 mm, and 2.23 degrees, respectively.

## 1. Introduction

Dental implants are a predictable treatment option for treating both partially and totally edentulous patients [1]. However, some complications may occur, causing implant failures. Clark et al. [2] showed that around 7% of complications might be related to implant malposition. Implants should be placed in the optimal position to reduce complications and maximise aesthetics. High implanting accuracy should be a major treatment goal [3,4,5]. 

To improve implanting accuracy, a dental implant navigation system (DINS) is proposed. The system is mainly composed of medical imaging, implanting instruments, optical positioning devices, and preoperational implant planning software [6,7]. The system performs real-time tracking of implanting instruments through optical markers and relates this information to the three-dimensional (3D) preoperative virtual plan drawn up with computed tomography (CT). Operators are guided to implant at the planned position; thus, highly accurate implant insertion can be acquired [8,9].

In the past decades, a large number of researchers have made their own contributions to the research of navigation systems. In 1986, Roberts et al. [10] developed the first surgical navigation system by integrating CT and ultrasonic positioning technology, and the system achieved high accuracy, good stability, and clinical effect. To further improve the positioning accuracy, image-guided surgery based on binocular stereo vision (BSV) was applied in the clinic [11]. With the development of dental CT, image-guided implant surgery was introduced into dental implantology to reduce deviations from the virtually planned implant position [12]. Siessegger et al. [13] proved that DINS was a valuable tool in implant surgery, and the image-guided techniques were superior to conventional implant techniques. In recent years, the visual surgery tracking platform and the robot operation platform have been integrated into DINS. Under the guidance of preoperative surgical planning, the robotic arm could automatically complete the preparation of implant holes to avoid human error [14,15]. For example, Yomi, approved by the Food and Drug Administration (FDA) in 2017, was the first commercial robot used for dental implants, and surgery has been successfully performed by a doctor holding this robot arm. Yuan et al. [15] designed a novel hybrid robot dedicated to dental implants. The hybrid robot independently completed the surgery under the guidance of the navigation system.

Dental implant navigation technology based on binocular stereo vision has been developed for decades. However, in actual operations, camera occlusion often leads to system interruption, which greatly improves the operation risk. Therefore, it is important to update the optical positioning technology. Trinocular stereo vision can shoot targets from multiple perspectives. When a camera is occluded, the system is still functional, providing a buffer time for doctors to make adjustments; thus, the robustness and security of the system can be significantly improved.

In this study, a dental implant navigation system based on trinocular stereo vision (TSV) is developed to overcome some disadvantages of the conventional method, and the accuracy of the system is evaluated in a phantom experiment.

The rest of this paper is organised as follows: in Section 2, the system setup and transformation relation of the coordinate system is described in detail. In Section 3, an implant hole preparation experiment is used to verify the feasibility and obtain the accuracy of the system. Then experimental results and discussion are demonstrated, followed by the conclusion given in Section 4.

## 2. System Setup and Principal

DINS mainly comprises implanting instruments, medical imaging, optical positioning devices, and preoperational implant planning software. An experimental DINS is constructed, as shown in Figure 1. In the system, each part has its own independent coordinate system. To guide operators to execute the implant surgery, it is important to unify all independently defined coordinate systems into one coordinate system. The relationship between hardware composition and the coordinate system of DINS is shown in Figure 2.

### 2.1. Trinocular Stereo Vision

TSV is used to detect the position of the patient and surgical instruments. TSV calibration and multi-view feature point matching should be prerequisites.

#### 2.1.1. Trinocular Stereo Vision Calibration

The 3D information of markers is obtained by TSV through the aerial triangulation method. The mathematical model of TSV is obtained by calibration. Compared with single camera calibration, TSV calibration aims to unify three cameras to the same coordinate system, called the world coordinate system. In this study, the world coordinate system is unified into the same coordinate system as the medium camera, whose origin has been set to its optical centre. The schematic diagram is shown in Figure 3.

Zhang’s camera calibration method is used to calibrate each camera [16]. The internal and external parameters and distortion coefficients of the camera can be obtained after calibration. According to the distortion coefficient, the distortion of the images is corrected. $R_{r}$,$\;T_{r}$,$\;R_{m}$,$\;T_{m}$,$\;R_{l}$,$\;T_{l}$ are the extrinsic parameters (rotation and translation) of the right, medium, and left cameras respectively. Equations are expressed as: 

Left camera:(1)$$\left[\begin{array}{c} x_{l}\\ y_{l}\\ z_{l} \end{array}\right]=R_{l}\left[\begin{array}{c} X_{w}\\ Y_{w}\\ Z_{w} \end{array}\right]+T_{l},$$

Medium camera:(2)$$\left[\begin{array}{c} x_{m}\\ y_{m}\\ z_{m} \end{array}\right]=R_{m}\left[\begin{array}{c} X_{w}\\ Y_{w}\\ Z_{w} \end{array}\right]+T_{m},$$

Right camera:(3)$$\left[\begin{array}{c} x_{r}\\ y_{r}\\ z_{r} \end{array}\right]=R_{r}\left[\begin{array}{c} X_{w}\\ Y_{w}\\ Z_{w} \end{array}\right]+T_{r},$$
where $\left(\begin{array}{ccc} X_{w} & Y_{w} & Z_{w} \end{array}\right)$ is the position of the marker in the world coordinate system. $\left(\begin{array}{ccc} x_{l} & y_{l} & z_{l} \end{array}\right)$, $\left(\begin{array}{ccc} x_{m} & y_{m} & z_{m} \end{array}\right)$, and $\;\left(\begin{array}{ccc} x_{r} & y_{r} & z_{r} \end{array}\right)$ are the positions of the same marker in the right, medium and left cameras’ coordinate systems respectively. Formulas (1) and (2) are then converted into the following representation:(4)$$\left[\begin{array}{c} x_{l}\\ y_{l}\\ z_{l} \end{array}\right]=R_{lm}\left[\begin{array}{c} x_{m}\\ y_{m}\\ z_{m} \end{array}\right]+T_{lm}.$$

Rotation ($R_{lm}$) and translation ($T_{lm}$) parameters of the left camera relative to the medium camera can be expressed as:(5)$$R_{lm}=R_{l}R_{m}^{-1}\;,\;T_{lm}=T_{l}-R_{l}R_{m}^{-1}T_{m}.$$

Similarly, rotation ($R_{rm}$) and translation ($T_{rm}$) parameters of the right camera relative to the medium camera can be expressed as:(6)$$R_{rm}=R_{r}R_{m}^{-1}\;,\;T_{rm}=T_{r}-R_{r}R_{m}^{-1}T_{m}$$

Thus, the coordinate systems of three different cameras are unified into one world coordinate system.

#### 2.1.2. Feature Point Matching

In dental implant navigation surgery, multiple templates are simultaneously tracked by TSV, and at least three optical markers are attached to each template. Three images are obtained synchronously, and marker matching among multi-view images is executed subsequently. Epipolar constraints [17,18], ordering and geometrical constraints [19], and circular coded targets [20,21] are generally introduced to match the markers for BSV. However, these techniques have their own limitations. Epipolar constraint is unable to determine correct correspondences when two or more markers are coplanar with two optical centres. Ordering and geometrical constraints depend on the marker’s location to identify. If part of the markers is blocked, the marker cannot be defined by uniqueness. There are also certain restrictions on the markers’ number and layout. The circular coded target, a central circular marker surrounded by a coded band, distinguishes the locating marker from the recognition. This technique increases the target area but is not able to improve positioning accuracy. In this study, a bipolar line constraint algorithm based on TSV is proposed to match the same markers from different image planes. The matching process is shown in Figure 4.

Fundamental matrices are calculated by the corresponding relation between the markers of the stereo image pairs. $F_{lm},F_{mr}\;\mathrm{and}\;F_{lr}$ represent the fundamental matrices of left-middle, middle-right, and left-right cameras, respectively. Optical marker $P_{1}$ is searched by traversing the left image, epipolar $l_{21}$, is calculated in medium image plane according to the following equation:(7)$$l_{21}=F_{lm}P_{1}.$$

$P_{2}$ can be searched in $l_{21}$, epipolar in the right image plane is calculated as follows:(8)$$l_{31}=F_{lr}P_{1},\;l_{32}=F_{mr}P_{2}.$$

According to $l_{2}$ and $l_{3}$, the only intersection can be worked out. Using it as a seed, the whole marker can be obtained by region growing algorithm, then $P_{3}$ is calculated by gray centroid method. $P_{3}$ can be mapped to the medium image, and the uniqueness of $P_{2}$ can be verified. At this point, markers are successfully matched at the left, middle, and right image planes. The mismatching problem can be fundamentally solved by the bipolar line constraint based on TSV.

### 2.2. Medical Image Data

With the development of imaging devices, such as CT, Cone Beam CT (CBCT) and magnetic resonance imaging (MRI), 3D image data of the dental cavity can be easily obtained. They make it possible to simulate a prophetically driven implant placement with specific software [6]. Preoperative imaging devices have their own pre-defined coordinate systems. Patients with a U-type locating tube in the mouth are scanned by an imaging device before surgery, then the imaging data is presented in Digital Imaging and Communications in Medicine (DICOM) format. The definition of the medical coordinate system is given in files. In the coordinate system, the ideal implant axis is designed and calibration markers are identified in image data as well. It should be noted that the model data of the jaw, ideal implant data, and the calibration markers are aligned in the same coordinate system.

### 2.3. Implant Instruments Calibration

In dental implant surgery, a variety of implant surgical instruments are used to drill holes in the edentulous area. In this study, a T-type template with three active optical markers is installed on a commonly used implant hand piece. Structures of the self-designed implant instrument are shown in Figure 5a.

Active optical markers are installed with light-emitting diodes (LEDs); thus, no extra illumination is required. The markers on the T-type template are positioned by TSV. Since the drill is our main concern, obtaining the position relationship between the drill and the T-type template (the calibration process) is crucial. In dental implant surgery, it is necessary to consider not only the position deviation between the actual implant point and the ideal implant point but also the deviation between the actual implant axis and the ideal implant axis. Endpoint calibration and axis calibration are available. The calibration of the drill is conducted for real-time displaying of the drill model on the upper computer. The calibration process of implant instruments not only plays a decisive role in the overall performance of surgery but also affects the interaction, real-time performance, and stability of the system.

#### 2.3.1. Endpoint Calibration

To improve the calibration accuracy and facilitate the operation, the tip of the instrument is designed as a ball head, and the winding point is a semi-spherical pit matching the ball head, as shown in Figure 6a. The implant instrument is rotated around a certain point to determine the coordinates of the endpoint in the world coordinate system and make sure that the markers are kept within the range of visual measurement. A total of N position coordinates are collected. The following equations can be obtained on the basis of the distance constraint relationship:(9)$$\|P_{ij}-P\|=R_{i},$$
where $P_{ij}$ is the position of marker point i at time j, $R_{i}$ is the distance between the marker 𝑖 and the endpoint i∗N constraint equations in total are available. $R_{i}$ is eliminated as follows:(10)$$2\left[\begin{array}{ccc} X_{ij}-X_{1j} & Y_{ij}-Y_{1j} & Z_{ij}-Z_{1j} \end{array}\right]P=X_{ij}{}^{2}+Y_{ij}{}^{2}+Z_{ij}{}^{2}-X_{1j}{}^{2}-Y_{1j}{}^{2}-Z_{1j}{}^{2}.$$

The equation is expressed as a matrix form:(11)GP=D,
where G is the 3 column $i\times \left(j-1\right)$ row matrix, D is $i\times \left(j-1\right)$ column matrix. The least squares method is applied to calculate the world coordinate  P:
(12)$$P={\left(G^{T}G\right)}^{-1}G^{T}D.$$

The rotation and translational matrices of the T-type template are calculated to obtain coordinates of the endpoint at any time. In the calibration process, the world coordinates of the markers at a certain time are stored; thus, the rotation and translation matrices at any time can be calculated by corresponding point matching. The coordinates of the endpoint at any time can be calculated by using:(13)$$P_{rtp}=R_{tg}P_{tp}+T_{tg},$$
where $R_{tg},T_{tg}$ are the rotation and translation matrices of the implant instrument. 

#### 2.3.2. Axis Calibration

After the endpoint calibration, the next step is to calibrate the axis of the drill, and obtain the vector of the drill axis in the world coordinate system. In the axis calibration process, systems follow the workflow as follows:(1)A short ball drill is installed on the self-designed implant instrument, and the effective length $l_{1}$ of the drill is recorded. The first calibration is completed by using the calibration method in Section 2.3.1. Recorded data include endpoint coordinate $P_{t1}\left(x_{1},y_{1},z_{1}\right)$ and marker coordinates $P_{01}\left(x_{1i},y_{1i},z_{1i}\right)$ at time $t_{1}$.(2)A long ball drill is installed on the self-designed implant instrument, and the effective length $l_{2}$ of the drill is recorded. The second calibration is also completed by using the calibration method in Section 2.3.1. Recorded data include endpoint coordinate $P_{t2}\left(x_{2},y_{2},z_{2}\right)$ and marker coordinates $P_{02}\left(x_{2i},y_{2i},z_{2i}\right)$ at time $t_{2}$.


The calculation process is shown in Figure 6b. The data in $t_{1}$ and $t_{2}$ is transformed into the same time. Since the T-type template is rigid, the rotation and translation matrices $R_{12}$ and $T_{12}$ from $t_{1}$ to $t_{2}$ can be obtained by using equations:(14)$$\mathrm{Min}\left\{\left|P_{02}-R_{12}P_{01}-T_{12}\right|\right\}.$$

$P_{t1}$ is transformed to time $t_{2}$, $P_{t1}^{’}\;$ is calculated as follows: (15)$$P_{t1}^{’}=R_{12}P_{t1}+T_{12}.$$

The vector of the axis can be calculated as:(16)$$\vec{v}=P_{t2}-P_{t1}^{’}.$$

The axis of the drill is calibrated, and the vector can be calculated at any time:(17)$$\vec{v_{r}}=R_{tg}\vec{v}+T_{tg}.$$

During dental implant surgery, once the surgical drill is replaced, the endpoint needs to be repositioned. The endpoint of the replacement drill can be calculated from the axis calibration results. The formula is as follows: (18)$$P_{t}=(P_{t2}-P_{t1}^{’})\frac{l_{3}-l_{1}}{l_{2}-l_{1}}+P_{t1}^{’},$$
where $l_{3}$ is the length of the replacement drill.

#### 2.3.3. Drill Calibration 

In dental implant surgery, the position and posture of implant surgical instruments relative to the edentulous area are displayed in real time. Because the surgical instrument is large, only the drill part is usually drawn for 3D display. However, TSV only locates optical markers on the T-type template, so the spatial transformation relationship between the implant drill and the T-type template needs to be established, as shown in Figure 5b.

To establish the transformation relationship between two coordinate systems, at least three non-collinear matching points are needed. Three calibration algorithms based on short, long, and bending ball drills are introduced in this study.

The bending ball drill, as shown in Figure 6c, is designed. The calibration of the bending drill is completed by using the calibration method in Section 2.3.1. Recorded data include: the endpoint coordinate $\;P_{t3}\left(x_{3},y_{3},z_{3}\right)$ and markers coordinates $P_{03}\left(x_{3i},y_{3i},z_{3i}\right)$ at time $t_{3}$ in the world coordinate system.

The rotation and translation matrices $R_{32}$ and $T_{32}$ from $t_{3}$ to $t_{2}$ can be obtained by Equations (14). $P_{t3}$ is transformed to time $t_{2}$. $P_{t3}^{’}\;$ is defined as follows:(19)$$P_{t}=(P_{t2}-P_{t1}^{’})\frac{l_{3}-l_{1}}{l_{2}-l_{1}}+P_{t1}^{’},$$

The endpoint coordinates of the short, long, and bending ball drills in world coordinate system are $P_{t1}^{’},\;P_{t2}\;\mathrm{and}\;P_{t3}^{’}$, respectively. Endpoint coordinates $P_{mt1},\;P_{mt2}\;\mathrm{and}\;P_{mt2}\;$ in model coordinate system have been identified in the design process. According to the corresponding point matching, the translation relationship ($R_{2},T_{2})$ of model coordinate system to world coordinate system is calculated. Coordinates of the optical markers on the T-type template at any time can be obtained by using TSV, then the rotation and translation matrices ($R_{tg},\;T_{tg}$) from $t_{2}$ to any time can be directly calculated, and coordinates of the model in world coordinate system at any time can be calculated by:(20)$$P_{rmt}=R_{tg}R_{2}P_{mt}+R_{tg}T_{tm}+T_{tg},$$
where $P_{rmt}$ is the coordinates of drill model in world coordinate system at any time and $P_{mt}$ is the coordinates of the drill model in model coordinate system.

The high-precision bending ball drill is designed and manufactured for three calibration algorithms. This increases the hardware cost, the operation difficulty, and the time of operation preparation. In dental implant surgery, the implant axis needs to be considered, but the freedom of rotation around the axis of the drill should not be limited. Therefore, the third point can be set artificially to calculate the conversion relationship between the drill model coordinate system and the world coordinate system.

The vectors of the drill axis in the model coordinate system and the world coordinate system are obtained separately by using:(21)$$\vec{v_{1}}=p_{mt2}-p_{mt1},\;\vec{v_{2}}=p_{t2}-p_{t1}^{’}.\;$$

The normal vector can be calculated by:(22)$$\vec{v_{3}}=\vec{v_{1}}\times \vec{v_{2}.}$$

The coordinates of the third matching point are obtained by extending the normal vector displacement unit distance with as the origin $p_{mt1}$ and $\;p_{t1}^{’}$, $p_{mt3}$ and $p_{t3}$ are calculated as follows:(23)$$p_{mt3}=p_{mt1}+\frac{\vec{v_{3}}}{\left|\vec{v_{3}}\right|},\;\;p_{t3}=p_{t1}^{’}+\frac{\vec{v_{3}}}{\left|\vec{v_{3}}\right|}.$$

According to $\left\{p_{mt1},p_{mt2},p_{mt3}\right\}$ and $\left\{p_{t1}^{’},p_{t2},p_{t3}\right\}$, the rotation and translation matrix ($R_{2},T_{2})$ is calculated. The calibration of drill is completed.

### 2.4. Reference Template

A series of optical markers are mounted on the reference template, and the template is connected to the jaw by connecting rod and clamping device. It can be considered that the relative position between the reference template and the jaw remains unchanged during the operation. The markers are reconstructed by TSV, then the real-time position information of the reference template can be calculated. Finally, the patient’s head movement process can be corrected to ensure that the 3D display of the jaw model remains static. 

After calibration and registration, the coordinates of the preoperative image and drill can be transformed to the same reference coordinate system, and then the drill and jaw model are 3D displayed on the computer screen. The conversion relationships of each coordinate system are shown in Figure 7.

## 3. Implant Hole Preparation Experiment and Result

Implant hole preparation experiments are carried out using self-designed DINS. The phantom jaws are used to explore the feasibility of the system and the precision of implant holes. The system setup is shown in Figure 8. Then, 850 nm near-infrared (NIR) filters are employed to eliminate ambient light interference, and three Basler AcA2000-165μmNIR CMOS cameras, NIR enhanced, are used for pleasurable image quality in the NIR band. The resolution (H x V) is 2048 pixels × 1088 pixels. As an NIR trinocular stereo vision system is constructed, the effective field of view is determined accordingly. The effective field is a maximum of 824 mm in the width direction, 437 mm in the short transverse direction, and 540–1380 mm in the depth direction.

In order to achieve spatial positioning, it is necessary to ensure the synchronisation of the three cameras. The external trigger mode is used in the system to synchronously capture image information. When the rising edge of an optical coupler input is received by cameras, a frame of the image is captured and cached in the internal buffer for subsequent processing. A 30 Hz frame rate is achieved by using a 30 Hz square wave.

The process of the dental implant hole preparation experiment is as follows:(1)CBCT data of dental is used to prepare a jaw model, along with the use of partly edentulous lower jaws with missing molars 46.(2)The U-type locating tube with development points is fixed in the model. The image data of the model are obtained by CBCT. The ideal planting axis was designed with self-designed software, and the coordinates of developing points in the image coordinate system are extracted, as shown in Figure 9a.(3)The jaw model is installed in the head phantom and is adjusted and fixed in the appropriate position. The reference template is clamped on the jaw model by a connecting rod, as shown in Figure 7, and the template is adjusted to the appropriate position and angle to suit the TSV measurement field of view.(4)The endpoint, axis, and drill are calculated based on the algorithm in Section 2.3. After calibration, the position and posture of the drill can be displayed in real time on the upper computer.(5)The markers on the U-type locating tube are clicked in a certain order to complete image registration.(6)The drill is replaced and the implant instrument is adjusted to drill under the guidance of the navigation software, as shown in Figure 10.(7)Boreholes are measured to obtain the deviation between the actual planting axis and the ideal planting axis.

The ideal implanting axis and the actual implanting axis are mapped to the same coordinate system by registration. The measured points on the axis of the borehole and the ideal implanting axis are shown in Figure 11. The red lines are the coordinates mapped from the ideal implanting axis to the world coordinate system, the green points are the actual measuring points of the boreholes, and the blue lines are the actual implanting axis fitted based on the measuring points. It can be seen from the figure that distances (less than 1 mm) between the actual implant axis and the ideal implant axis meet the demands of dental implants.

The accuracy of DINS was evaluated utilising the entry, exit, and angle deviation between the implants and the planned trajectories [22], as shown in Figure 9c. Ten groups of repetitive experiments were conducted, and the results are shown in Table 1. It can be seen from the table that the mean deviation and standard deviation of the entry point are 0.553 mm and 0.203 mm, the mean deviation and standard deviation of the exit point are 0.878 mm and 0.315 mm, and the mean deviation and standard deviation of the angle are 2.23 degrees and 0.989 degrees. The accuracy of the DINS is satisfying for dental implants. 

To evaluate the advancement of implant method provided in this study, the results of some existing methods from other researchers’ literatures are listed in Table 2. It can be seen that there are significant differences for all deviation parameters found in the results of implant-guided placement, compared to placement without guidance, and the proposed method in this study is shown to be better than the other methods. In general, DINS based on TSV can meet the requirements of clinical conditions and accuracy [23]. 

## 4. Discussion and Conclusions

In this study, DINS based on TSV is constructed. Compared with the traditional optical navigation system based on BSV, TSV provides stronger anti-occlusion ability and higher positioning accuracy. The feasibility of the system for dental implantation is verified by an implant hole preparation experiment.

In terms of instrument calibration, drills with ball heads are designed to improve calibration accuracy, and they rotate around the spherical pit to facilitate calibration operation. Two calibration algorithms based on short and long ball drills are proposed to register the axis and drill instead of high-precision axis calibration devices, then the operation steps are simplified and the system cost is reduced.

The accuracy of the systems is investigated by using phantom jaws. Concerning the experimental results, the mean entry deviation, exit deviation, and angle deviation are 0.55 mm, 0.88 mm, and 2.23 degrees, respectively, which meet the requirement of dental implants. Figure 11a–d show that there are overall deviations of the sub-millimetre between the ideal implanting axis and the actual implanting axis. The relevant TSV positioning experiment shows that the single point positioning accuracy can reach 0.024 mm [27], and the accuracy is far better than implant accuracy in this study. Errors mainly come from the following three parts. First, image data errors. In the process of CBCT data acquisition and processing, sub-millimetre error may occur [28]. For example, the thickness of the CBCT scanning layer in this study is 0.2 mm. Second, calibration, and registration errors. Coordinate systems need to be transformed many times in this system, and calculation and operation errors are inevitable in every coordinate transformation. These errors eventually accumulate in the screen coordinate system and affect dental implant guidance. Third, operators-related errors [29,30]. The operator handles surgical instruments with the guidance of the DINS. Theoretically, the endpoint should be close to the pre-planned path, and the accuracy should be high enough. However, due to tremor and other reasons, when the deviation between the endpoint and the ideal implanting axis is less than 0.2 mm, it is difficult for the operator to further reduce the operation error in a short period of time. Furthermore, the jitter of drill also affects the implanting accuracy. 

Reducing the errors in the above aspects needs further study. In addition, phantom experiments cannot completely simulate the real clinical situation. Animal and even cadaver experiments should be conducted in the future.

## Figures and Tables

**Figure 1 sensors-22-02571-f001:**
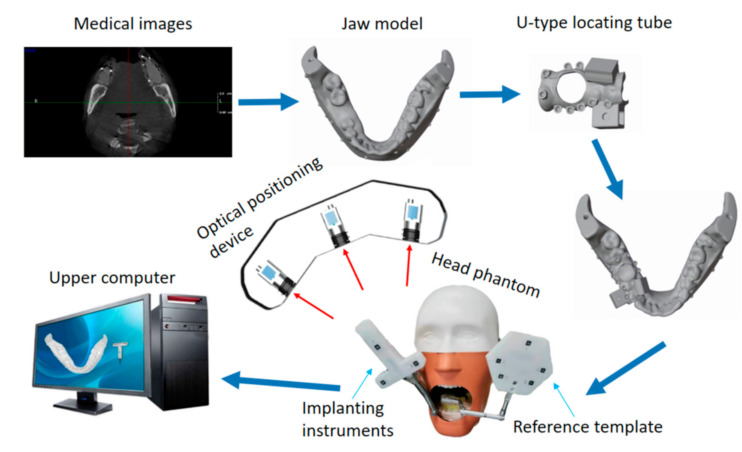
The scheme of the dental implant navigation system (DINS) is based on trinocular stereo vision (TSV).

**Figure 2 sensors-22-02571-f002:**
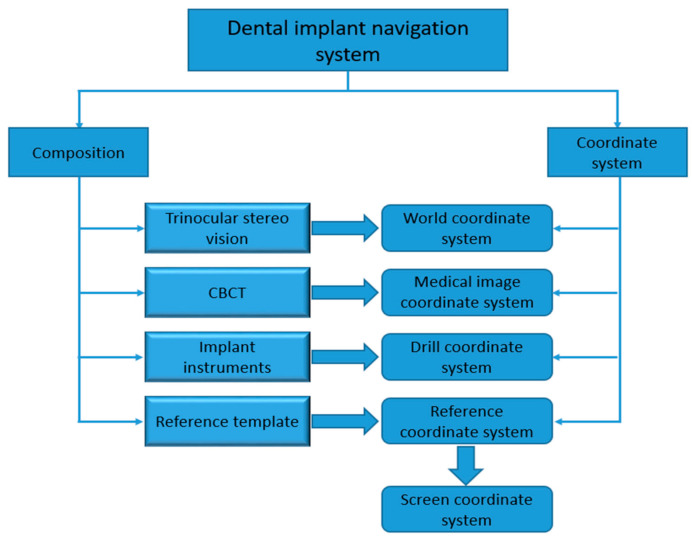
The relationship between hardware composition and the coordinate system of DINS.

**Figure 3 sensors-22-02571-f003:**
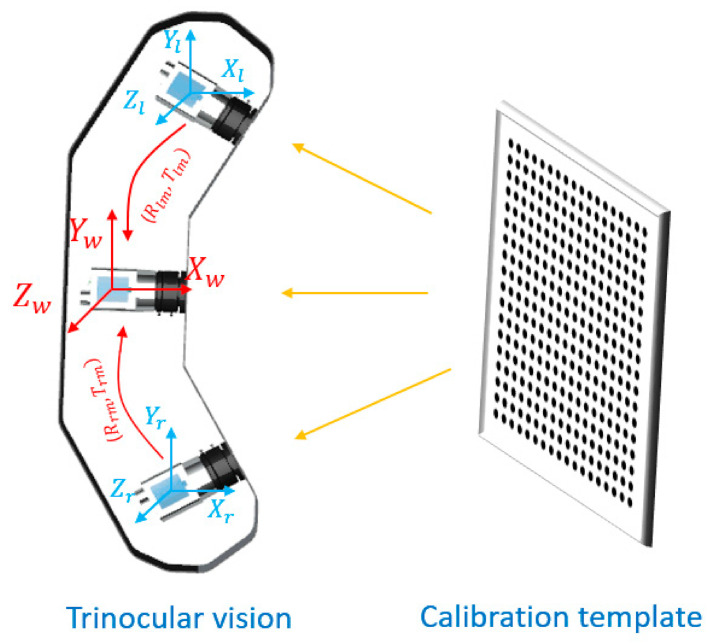
Calibration of TSV.

**Figure 4 sensors-22-02571-f004:**
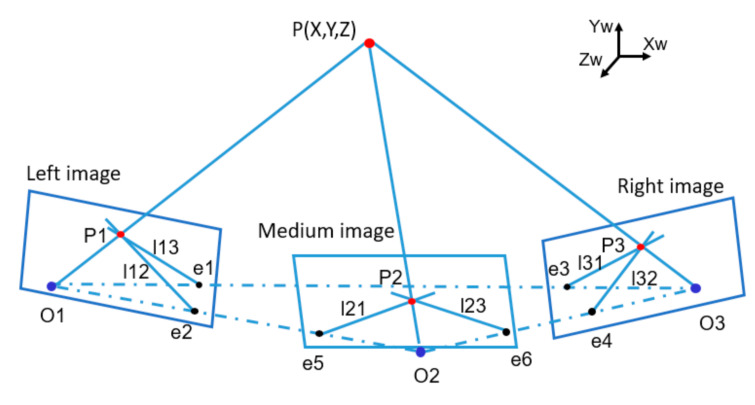
Schematic diagram of bipolar line constraints in TSV.

**Figure 5 sensors-22-02571-f005:**
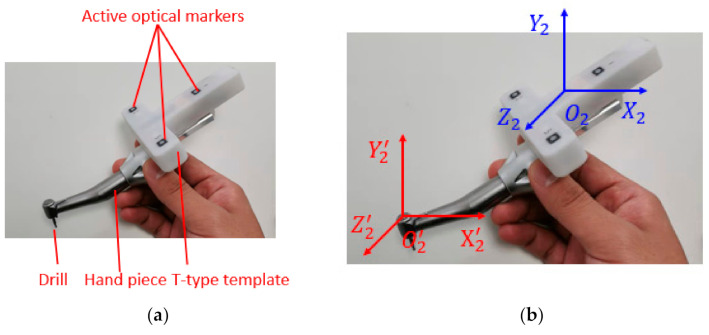
Self-designed implant instrument (**a**). Component, (**b**) coordinate system transformation.

**Figure 6 sensors-22-02571-f006:**
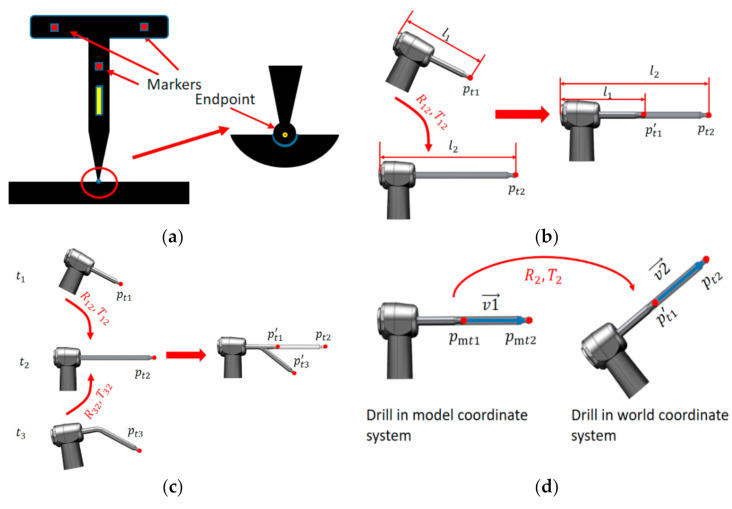
Schematic diagram of implant surgical instrument calibration (**a**). Endpoint calibration, (**b**) axis calibration algorithm based on short and long ball drills, (**c**) three calibration algorithms based on short, long, and bending ball drills, and (**d**) two calibration algorithms based on short and long ball drills.

**Figure 7 sensors-22-02571-f007:**
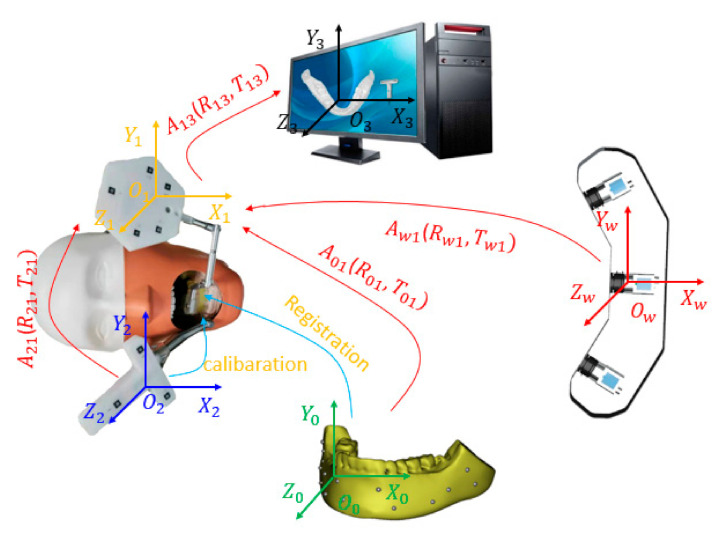
The coordinate transformation of the system,$\;O_{w}$ is the world coordinate system, $O_{o}$ is the medical image coordinate system, $O_{1}$ is the reference coordinate system, $O_{2}$ is the drill coordinate system, and $O_{3}$ is the screen coordinate system.

**Figure 8 sensors-22-02571-f008:**
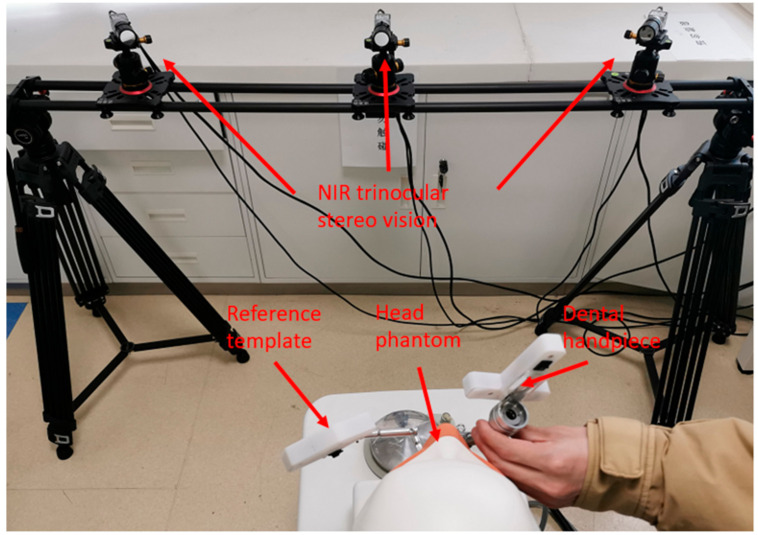
System setup.

**Figure 9 sensors-22-02571-f009:**
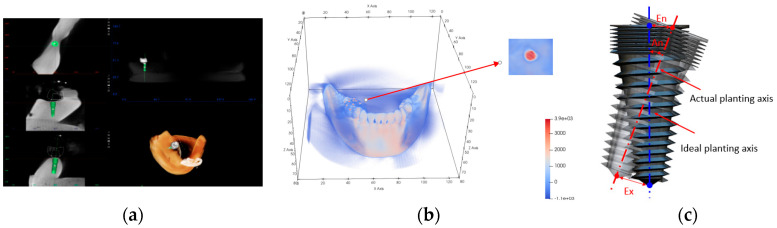
Image data processing. (**a**) Ideal implant is designed, (**b**) positioning markers are extracted, (**c**) comparison between the actual trajectory and the planned trajectory, An represents angle deviation, En represents entry deviation, Ex represents exit deviation.

**Figure 10 sensors-22-02571-f010:**
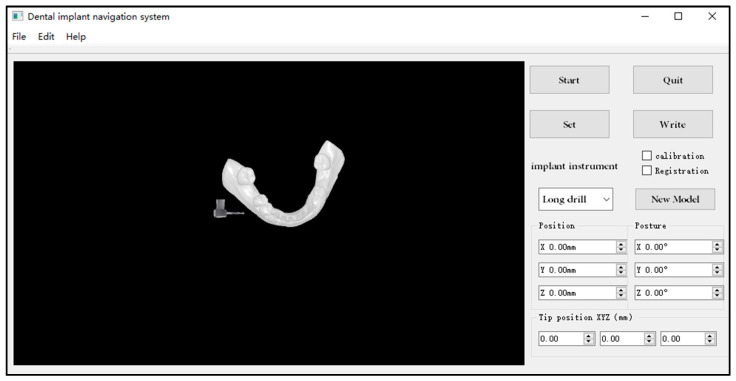
The dynamic real-time representation on the computer.

**Figure 11 sensors-22-02571-f011:**
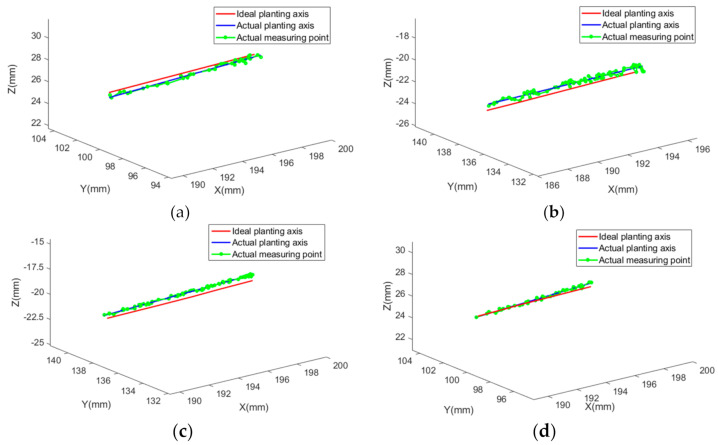
Deviation diagram between ideal implanting axis and actual implanting axis. (**a**–**d**) shows that the actual planting axis is around the ideal planting axis, and distributions of different tests are different. A deviation between the ideal planting axis and the actual planting axis is mainly composed of the calibration and registration error of the navigation system, therefore, the deviation is not fixed.

**Table 1 sensors-22-02571-t001:** Resulting deviations for the system.

Groups	Entry Deviation (mm)	Exit Deviation (mm)	Angle Deviation (degree)
1	0.459	0.734	1.58
2	0.561	1.070	2.91
3	0.825	1.447	3.56
4	0.149	0.561	2.36
5	0.731	0.578	0.88
6	0.614	0.813	1.14
7	0.760	1.404	3.68
8	0.312	0.803	2.81
9	0.432	0.859	2.45
10	0.682	0.514	0.96
Mean	0.553	0.878	2.23
Std	0.203	0.315	0.989

**Table 2 sensors-22-02571-t002:** Comparison of results obtained from different implant methods. “Proposed” represents the system proposed in this paper.

Method	Entry Deviation (mm)	Exit Deviation (mm)	Angle Deviation (degree)
Manual operation [11]	1.67	2.51	7.69
Block et al. [24]	1.37	1.56	3.62
Tahmaseb et al. [25]	1.45	2.99	4.00
Stefanelli et al. [26]	0.71	1.00	2.26
Proposed	0.55	0.88	2.23

## Data Availability

Not applicable.

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
