# Peer review of "Dental Implant Navigation System Based on Trinocular Stereo Vision"

_sensors, 2022, doi:10.3390/s22072571_

Round 1

Reviewer 1 Report

The authors present a three-camera system for tracking the implanting instruments during dental operations. Though the results might be interesting to medical community, manuscript lacks important technical details.

1. What cameras, lenses and light source do you use? Why is the system called "NIR" in Figure 7? Does it really operate in near infrared range?

2. Joint flowchart of the calibration, data processing and measurement algoritms is necessary.

3. Image quality issues should be addressed and discussed. What is the image resolution, distortion?

4. Despite three cameras, the obtained error is just a bit lower than in the stereoscopic system presented in [26]. Seems like adding a camera should provide better measurements. Please comment on the metrological advntages of the proposed system.

Reviewer 2 Report

The article describes a navigation system for dental implantation that utilizes trinocular stereo vision, rather than the binocular stereo vision that has up to now become standard.  This is a reasonably-motivated development, since one (or even both) of the cameras used in binocular stereo vision can easily become blocked by either implantation tools or anatomical features at various points during implantation procedures.

The article should be published.  However, it could greatly use a thorough re-writing by a native English speaker.  For example, just in the first two sentences of the abstract of the article:

"Traditional dental implant navigation system (DINS) based on binocular stereo vision (BSV) have some limitations. For example, weak anti-occlusion ability and feature point mis-matching problem."

should instead be:

"Traditional dental implant navigation systems (DINS) based on binocular stereo vision (BSV) have limitations; for example: weak anti-occlusion abilities, as well as problems with feature point mismatching."

Also, in Section 4 "Discussion and conclusion", it appears to me that the first paragraph: "Authors should discuss the results and how they can be interpreted from the perspective of previous studies and of the working hypotheses. The findings and their implications should be discussed in the broadest context possible. Future research directions may also be highlighted." -- was just accidentally copied from a template article!!!

In any case, as a reviewer I can't exhaustively list all the myriad English fixes that are needed throughout the paper, but a thorough and careful update with such English fixes is needed.

As for the scientific content of the paper: it looks fine to me.  The only significant question in my mind is whether the trinocular stereo vision could/would benefit from having the focal axis of the third camera be _outside_ of the plane that is formed by the focal axes of the first two cameras.  It seems to me that such a placement (rather than having all three cameras being within the same plane, as shown in the article) could potentially improve depth perception.  Whether that is true or not -- and very preferably with some experimental justification -- would be very useful to have in this paper.

Congratulations to the authors for their hard work, and for a decent paper draft!

Round 2

Reviewer 1 Report

After revision, the paper became more informative and convincing. Technical side of this study still needs a more detailed description.

1. What illumination (type, power, spectral range, etc) do you use? Do cameras have NIR filters?

2. What about data acquisition and processing time? What are the framerates of the cameras? How did you synchronize all three cameras?

3. When you speak about Zhang's camera calibration method, why do you refer to [16]? It makes sense to include a reference to initial Zhang's paper.
